# STAR: NEXT-SCALED AUTOREGRESSIVE MODEL FOR TIME SERIES FORECASTING

## ABSTRACT

We present the Next-Scaled Time-Series Autoregressive Model (STAR), a novel and effective method for time series forecasting that captures both the global structure and local details of temporal data. The architecture of STAR consists of two core components that work in tandem to improve the accuracy of the forecast. First, a transformer-based module models the coarse, long-term dynamics of the target series at a reduced scale, effectively capturing extended temporal dependencies. Second, a next-scale autoregressive module progressively refines forecasts from coarse to fine scales. It iteratively improves predictions using information from the preceding coarser scale, enabling precise reconstruction of fine-grained temporal dynamics. We conducted extensive experiments on seven widely used benchmark datasets in the time series forecasting domain. The results consistently demonstrate that STAR achieves state-of-the-art performance, significantly outperforming existing diffusion-based and transformer-based forecasting models across multiple evaluation metrics. Our code is available at `https://anonymous.4open.science/r/STAR-TSF/`.

## 1 INTRODUCTION

Time series forecasting is a crucial task across various fields, including finance Yun et al. (2023), healthcare Matsubara et al. (2014), weather prediction Wu et al. (2023), and energy consumption Martín et al. (2010), where accurate predictions of future trends are essential for decision-making and resource planning. Traditional forecasting models often struggle to model both the long-term dependencies and fine-grained details of complex temporal data.

Recent advances in deep learning have led to the development of more sophisticated models, yet existing methods still face challenges in capturing the full complexity of time series data, particularly in terms of effectively balancing global trends and local fluctuations. Traditional forecasting models often specialise in one of these aspects: for instance, transformer-based models Liu et al. (2023a); Wu et al. (2021); Liu et al. (2023b) and linear-based models Nie et al. (2024); Zeng et al. (2023) are well-regarded for their ability to capture global structures and long-range temporal patterns, making them effective for long-horizon forecasting. In contrast, progressive generative-based approaches such as diffusion-based models Rasul et al. (2021); Kollovieh et al. (2023); Gao et al. (2025) excel at modelling fine-grained short-term fluctuations. However, existing approaches typically struggle to strike a balance between these two objectives.

In this paper, we propose the Next-Scaled Time-series AutoRegressive Model (STAR), a forecasting method grounded in a coarse-to-fine prediction paradigm. The central idea is to first approximate the global shape of the time series at a reduced resolution and then progressively refine this representation to recover fine-grained temporal dynamics (Figure 1a). To operationalize this principle, STAR incorporates two complementary modules: the **Global-Shape Forecaster (GSF)** and the **Next-Scaled Refiner (NSR)**.

**The Global-Shape Forecaster** operates at a coarse scale in the token embedding space, capturing long-range structures such as periodicity, seasonality, and directional trends. Its forecasts establish a coherent and stable global foundation.

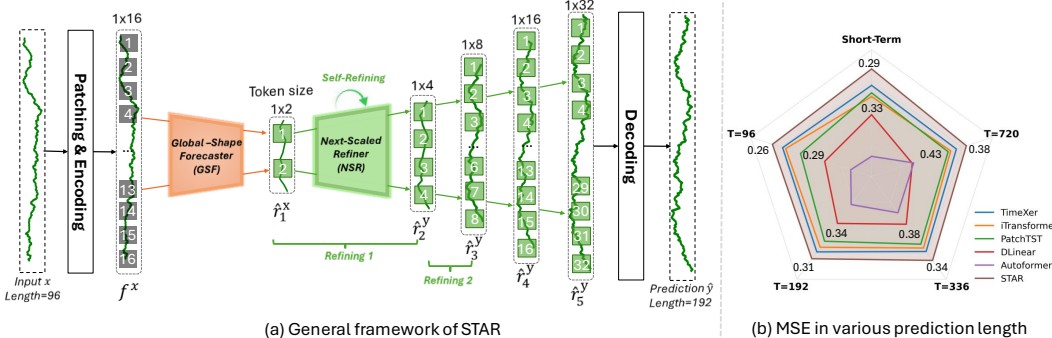

(a) General framework of STAR (b) MSE in various prediction length

Figure 1: (a) General architecture of the proposed STAR. The input time series is first divided into patches and encoded into patch tokens, each representing a sequence of consecutive timestamps. The Global-Shape Forecaster predicts a coarse initial representation of the target series, which is progressively refined by the Next-Scaled Refiner to reconstruct the full-resolution ground truth. (b) Performance comparison of STAR with state-of-the-art methods across diverse time series forecasting settings.

**The Next-Scaled Refiner** builds upon this foundation through a hierarchical refinement process. By progressively increasing the temporal resolution, it recovers localized variations that are often overlooked by global-only approaches.

However, training a next-scaled autoregressive model such as **NSR** is challenging due to the accumulation of errors across generation steps, which can cause output sequences to diverge significantly from the target patterns. This issue becomes even more pronounced as the model scales, amplifying even minor inaccuracies over time. To address this, we propose a technique called **Inference-Aware Augmentation (IAAug)**. IAAug first generates a more realistic input by running the model on the original input for several steps, simulating inference-stage behaviour. We then augment the original input by interpolating it with the generated input, improving the model's stability and reducing cumulative errors.

Leveraging these complementary components, STAR delivers high forecasting accuracy while maintaining both long-term coherence and short-term precision. As shown in Figure 1b, it consistently outperforms existing state-of-the-art (SOTA) models. Our main contributions are as follows:

- We propose **STAR**, an innovative time series forecasting method that combines the **Global-Shape Forecaster (GSF)** and the **Next-Scaled Refiner (NSR)** to address the dual challenges of capturing long-term trends and modelling local fluctuations, offering a comprehensive solution to complex temporal data.
- We introduce **Inference-Aware Augmentation (IAAug)**, a novel technique that minimises cumulative generation errors by generating more realistic input sequences and progressively refining forecasts.
- Through extensive experiments, we demonstrate that **STAR** achieves SOTA performance across a variety of forecasting tasks.

## 2 RELATED WORKS

**Time Series Forecasting.** Several approaches have been proposed for time series forecasting (TSF). Classical statistical methods such as ARMA Box (2013) and ARIMA Ariyo et al. (2014) model linear dependencies between past and present observations but often fall short in capturing the nonlinear patterns. Recent advances leverage deep learning, especially Transformer-based models, to capture long-range dependencies. Informer Zhou et al. (2021), Autoformer Wu et al. (2021), and others Zhang & Yan (2023); Liu et al. (a) apply attention mechanisms to temporal data. PatchTST Liu et al. (2023a) segments time series into tokens for attention-based learning, while iTransformer Liu et al. (2023b) attends across transposed dimensions to model multivariate relationships. Meanwhile, efficient linear models such as DLinear Zeng et al. (2023), RLinear Zeng & et al. (2023), TimeMixer Wang et al. (2024), and TimeXer Nie et al. (2024) perform well on large-scale datasets. LLM-based methods Liu et al. (2024); Tan et al. (2024) have also emerged, leveraging pre-trained Transformers and external data sources. Diffusion-based models, on the other hand, have also gained attention for their generative capabilities. TimeGrad Rasul et al. (2021) adopts an autoregressive strategy prone

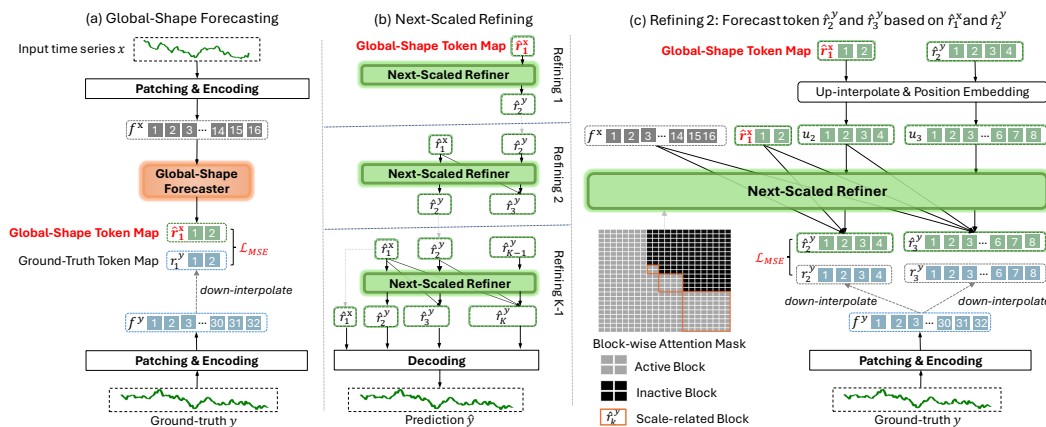

Figure 2: Overall architecture of the proposed STAR model. (a) The Global-Shape Forecaster generates the coarse prediction $\hat{r}_1^{\mathbf{x}}$; (b) the Next-Scale Refiner progressively refines $\hat{r}_1^{\mathbf{x}}$ through $K-1$ refining stages; (c) detailed pipeline of Refining Stage 2 of the Next-Scale Refiner, $u_k$ means the up-interpolated version of $\hat{r}_{k-1}^{\mathbf{y}}$.

to long-term error accumulation. In contrast, D3VAE Li et al. (2022) improves accuracy through probabilistic modelling and multi-scale denoising, while TSDiff Kollovieh et al. (2023) introduces self-guided inference without modifying the training pipeline. ARMD Gao et al. (2025) forecasts future segments directly via an autoregressive moving strategy. These models perform well on short-term tasks, thanks to their detailed multi-step generation capabilities. While Transformer-based and linear models excel at modelling global trends and long-term dependencies, they often struggle with fine-grained local details. Diffusion-based models, in contrast, capture local patterns effectively but face challenges in maintaining global consistency.

**Coarse-to-Fine AutoRegressive Model.** Several methods have recently demonstrated the effectiveness of a small-to-large generation strategy (Ye et al., 2024; Sahoo et al., 2024; Tian et al., 2024). For instance, MDLM (Sahoo et al., 2024) employs a progressive masking-and-filling mechanism to gradually complete text sequences, thereby enhancing long-range consistency in language modeling. In the visual domain, VAR (Tian et al., 2024) adopts next-scale generation, first producing coarse, low-resolution images and then refining them to full resolution. In video generation, models such as Make-A-Video (Singer et al., 2022) follow a similar principle, generating temporally coarse sequences before refining both spatial and temporal fidelity. Likewise, in speech synthesis, models like FastSpeech (Ren et al., 2019) first predict coarse phoneme-level structures and subsequently refine them into high-fidelity waveforms.

## 3 PROPOSED METHOD

We propose the Next-Scaled Time Series AutoRegressive (STAR), a forecasting method grounded in a coarse-to-fine prediction paradigm. The central idea is to first approximate the global shape of the time series at a reduced resolution and then progressively refine this representation to recover fine-grained temporal dynamics (Figure 1a).

To operationalize this principle, we first introduce the **Multi-Scale Time Series Autoencoder** (Section 3.2), which encodes and decodes the target time series into multi-scale tokens representing small- to large-scale temporal structures. Next, we present the **Global-Shape Forecaster**, which leverages a standard transformer to forecast the initial coarse token from the historical time series, capturing long-term dependencies and global patterns. The **Next-Scaled Refiner**, described in Section 3.4, progressively refines predictions from smaller to larger scales, get the more detail information. Section 3.4 also introduces **Inference-Aware Augmentation** (IAAug), a technique that improves model stability by mitigating cumulative errors during multi-step generation. Finally, the overall STAR architecture is summarized in Section 3.5 and illustrated in Figure 2.

### 3.1 PRELIMINARY

**Time Series Forecasting (TSF):** Given a historical time series $\mathbf{x} = \{x_1^1, \ldots, x_L^C\} \in \mathbb{R}^{C \times L}$, where $C$ is the number of variables and $L$ is the sequence length, the goal of TSF is to predict future values

$\mathbf{y} \in \mathbb{R}^{C \times T}$ over the next $T$ time steps. When $C = 1$, the problem is univariate, while when $C > 1$, it becomes multivariate.

**Time Series Patching:** To capture local temporal patterns, we divide each channel of the time series $\mathbf{y} \in \mathbb{R}^{C \times T}$ into $P$ (optionally overlapping) patches of length $T/P$. This produces a patch representation $\mathbf{z} \in \mathbb{R}^{C \times P \times T/P}$.

## 3.2 MULTI-SCALE TIME SERIES AUTOENCODER

Given the future time series $\mathbf{y} \in \mathbb{R}^T$ and its patching feature $\mathbf{z} \in \mathbb{R}^{C \times P \times T/P}$, we embed $\mathbf{z}$ using encoder $\mathcal{E}$ to get the feature embedding $f \in \mathbb{R}^{C \times P \times D}$, where D is the embedding size.

To process the next-scaled prediction with $K$ step, we firstly propose a **Multi-Scale Time Series Autoencoder**, which embeds the future time series $\mathbf{y}$ into a meaningful multi-scale token map $R = \{r_1, r_2, ..., r_K\}$, where $r_k \in \mathbb{R}^{C \times P_k \times D}$ is the $k^{th}$ token map in which $P_k$ is token map size at step $k$ with $P_K = P$.

**Encoding (Algorithm 1).** Given the token map size list $\{P_k\}_{k=1}^K$, the feature map $f$ is further processed through down-interpolation (reducing token size), up-interpolation (increasing token size), and residual quantization. This transforms $f$ into a sequence of multi-scale token maps $R = \{r_1, r_2, ..., r_K\}$, where $r_k \in \mathbb{R}^{C \times P_k \times D}$ is the $k^{th}$ token map with $P_k$ as the $k^{th}$ token map size.

$$R, f = \text{Encoding}(\mathbf{y}). \tag{1}$$

Unlike independent interpolation, we adopt a residual-style, where each $r_k$ is conditioned on its predecessors $\{r_1, r_2, \ldots, r_{k-1}\}$. This sequential dependency promotes consistency across scales and enables earlier (coarser) tokens to guide the generation of finer ones. To mitigate information loss during up-interpolating, we introduce a set of $K$ lightweight layers $\{\phi_k\}_{k=1}^K$. These layers refine the upsampled features at each scale to recover fine-grained details.

**Decoding (Algorithm 2).** The decoding procedure reconstructs $\hat{\mathbf{y}}$ from the input time series from the sequences of token maps $R$. The decoder $\mathcal{D}$ takes the hierarchical token maps and synthesises the feature map $\hat{f}$, which is then decoded into the time series $\hat{\mathbf{y}}$. The convolutional layers $\{\phi_k\}_{k=1}^K$ are also applied during decoding to ensure high-fidelity reconstruction at each specific length.

$$\{\hat{\mathbf{y}}_1, ..., \hat{\mathbf{y}}_K\}, \hat{f} = \text{Decoding}(R). \tag{2}$$

**Training Objective.** We jointly optimise the encoder $\mathcal{E}$, decoder $\mathcal{D}$, and convolutional refiners $\{\phi_k\}_{k=1}^K$ using a combination of reconstruction and adversarial losses:

$$\mathcal{L}_{\text{ae}} = \frac{1}{K} \sum_{k=1}^K \mathcal{L}_{\text{mse}}(\hat{\mathbf{y}}_k, \mathbf{y}_k) + \mathcal{L}_{\text{mse}}(f, \hat{f}) + \mathcal{L}_{\text{gan}}(\hat{\mathbf{y}}). \tag{3}$$

Here, $\mathcal{L}_{\text{mse}}$ measures the reconstruction fidelity in both data and feature space, while $\mathcal{L}_{\text{gan}}$ introduces generative regularization, encouraging the model to produce more realistic outputs, especially when modeling high-frequency variations. The GAN loss can be instantiated using techniques from the time series GAN literature (Goodfellow et al., 2014).

---

**Algorithm 1:** Encoding

**Input:** Time series $\mathbf{y}$, Token map size list $\{P_k\}_{k=1}^K$.
1   $f_{\text{init}} = f = \mathcal{E}(\text{Patching}(\mathbf{y}))$;
2   **for** $k = 1...K$ **do**
3      $r_k = \text{down-interpolate}(f, P_k)$;
4      $R.\text{append}(r_k)$;
5      $r_k = \text{up-interpolate}(r_k, P_K)$;
6      $f = f - \phi_k(r_k)$;
7   **return** $R, f_{\text{init}}$

---

**Algorithm 2:** Decoding

**Input:** Token map $R$, Token map size list $\{P_k\}_{k=1}^K$.
1   $\hat{f} = 0, \{\hat{\mathbf{y}}_1 = 0, ..., \hat{\mathbf{y}}_K = 0\}$;
2   **for** $k = 1...K$ **do**
3      $r_k = R[k]$;
4      $r_k = \text{up-interpolate}(r_k, P_K)$;
5      $\hat{f} = \hat{f} + \phi_k(r_k)$;
6      $\hat{\mathbf{y}}_k = \mathcal{D}(\hat{f})$;
7   **return** $\{\hat{\mathbf{y}}_1, ..., \hat{\mathbf{y}}_K\}, \hat{f}$

---

To ensure that each token can present the coarser time series, we compute the loss of each pair $(\hat{\mathbf{y}}_k, \mathbf{y}_k)$ where $\hat{\mathbf{y}}_k$ is the output time series decoded by $k^{th}$ token map and $\mathbf{y}_k$ is the coarser time series of ground-truth $\mathbf{y}$ in step $k$.

$$\mathbf{y_k} = \text{up-interpolate}(\text{down-interpolate}(\mathbf{y}, T_k), T_K), \tag{4}$$

where $T_1, \ldots, T_K$ denotes the length of the time series at each generation step, with $T_K = T$. Note that $T_k$ and $P_k$ share the same scaling factor, i.e., $\frac{T_k}{T} = \frac{P_k}{P}$. Consequently, $\mathbf{y}_k$ represents a coarser version of the ground-truth time series, scaled by the factor $\frac{T_k}{T}$.

### 3.3 Global-Shape Forecaster

The Global-Shape Forecaster (GSF) is designed to model long-term dependencies and capture high-level temporal patterns, such as trends and seasonality. For simplicity, we adopt a standard transformer $\mathcal{T}$ as the architecture for GSF. Firstly, we will extract the feature of time series using encoder $\mathcal{E}$.

$$f^{\mathbf{x}} = \mathcal{E}(\text{Patching}(\mathbf{x})). \tag{5}$$

Then, the transformer $\mathcal{T}$ processes $f^{\mathbf{x}}$ to predict the first global-shape token map $\hat{r}_1^{\mathbf{x}}$.

$$\hat{r}_1^{\mathbf{x}} = \mathcal{T}(f^{\mathbf{x}}). \tag{6}$$

For the target sequence $\mathbf{y}$, we compute the ground-truth feature map and sequences of token maps $R^{\mathbf{y}}, f^{\mathbf{y}} = \text{Encoding}(\mathbf{y})$ (Algorithm 1) and extract the first ground-truth token maps $r_1^{\mathbf{y}} = R^{\mathbf{y}}[1]$. The training objective minimizes the difference between the predicted and ground-truth maps:

$$\mathcal{L}_{\mathcal{T}} = \mathcal{L}_{\text{mse}}(\hat{r}_1^{\mathbf{x}}, r_1^{\mathbf{y}}). \tag{7}$$

### 3.4 Next-Scaled Refiner

The **Next-Scaled Refiner (NSR)** progressively refines predictions across multiple lengths by autoregressively generating prediction token maps from coarse to fine scales. Specifically, given historical and future time series $\mathbf{x}$ and $\mathbf{y}$, our objective is to forecast the multi-scale token maps $\{r_1^{\mathbf{y}}, ..., r_K^{\mathbf{y}}\}$ of $\mathbf{y}$ from coarse to fine scales conditioned on $\mathbf{x}$. Each token map $r_k^{\mathbf{y}} \in \mathbb{R}^{C \times P_k \times D}$ is conditioned on its prefix sequence $(f^{\mathbf{x}}, r_1^{\mathbf{y}}, r_2^{\mathbf{y}}, ..., r_{k-1}^{\mathbf{y}})$ and position embeddings. At each scale, all tokens in $r_k^{\mathbf{y}}$ are predicted in parallel, enabling efficient refinement:

$$p(r_1^{\mathbf{y}}, \ldots, r_K^{\mathbf{y}}) = \prod_{k=2}^{K} p(r_k^{\mathbf{y}} \mid f^{\mathbf{x}}, r_1^{\mathbf{y}}, ..., r_{k-1}^{\mathbf{y}}). \tag{8}$$

Given NSR autoregressive backbone $\mathcal{A}$, we denote the generation process in STAR as:

$$\hat{r}_2^{\mathbf{y}}, ..., \hat{r}_K^{\mathbf{y}} = \mathcal{A}(\hat{r}_1^{\mathbf{x}}, r_2^{\mathbf{y}}, ..., r_{K-1}^{\mathbf{y}}), \tag{9}$$

where we replace $(f^{\mathbf{x}}, r_1^{\mathbf{y}})$ by $\hat{r}_1^{\mathbf{x}}$ (cf. (7)) and we minimize the MSE loss between the outputs $\{\hat{r}_2^{\mathbf{y}}, ..., \hat{r}_K^{\mathbf{y}}\}$ of the autoregressive $\mathcal{A}$ and the ground-truth $\{r_2^{\mathbf{y}}, ..., r_K^{\mathbf{y}}\}$ to train $\mathcal{A}$.

**Block-wise Attention Mask.** To ensure that $\hat{r}_k^{\mathbf{y}}$ only utilizes information from the previous token sequence $(f^{\mathbf{x}}, \hat{r}_1^{\mathbf{x}}, r_2^{\mathbf{y}}, \ldots, r_{k-1}^{\mathbf{y}})$, we propose a block-wise causal attention mask, which restricts attention to preceding blocks. An example of our attention mask is shown in Figure 2c.

To better suit time series forecasting, STAR introduces two additional conditioning signals: the historical embedding $f^{\mathbf{x}}$ and the global-shape token map $\hat{r}_1^{\mathbf{x}}$. Whereas prior autoregressive models (Tian et al., 2024; Sahoo et al., 2024) primarily rely on previously generated tokens, STAR augments the autoregressive context with information-rich representations, enabling more effective guidance during generation.

More specifically, the historical embedding $f^{\mathbf{x}}$ captures contextual features from the input sequence $\mathbf{x}$, while the global-shape token map $\hat{r}_1^{\mathbf{x}}$, produced by the GSF, encodes long-term structural patterns such as trends and seasonality. These components provide a strong prior that improves training stability and predictive accuracy. As a result, STAR not only converges faster, but also generates more coherent and reliable forecasts for the future sequence $\mathbf{y}$ conditioned on the historical input $\mathbf{x}$.

**Training Procedure for NSR and GSF.** The pseudocode of this training pipeline is presented in Algorithm 3. During training, the historical input $\mathbf{x}$ and future target $\mathbf{y}$ are encoded using the encoder to obtain their respective feature maps and a sequence of multi-scale token maps: $f^{\mathbf{x}}, R^{\mathbf{x}} =$ Encoding$(\mathbf{x})$ and $f^{\mathbf{y}}, R^{\mathbf{y}} =$ Encoding$(\mathbf{y})$. The GSF backbone $\mathcal{T}$ generates the global-shape token map $\hat{r}_1^{\mathbf{x}} = \mathcal{T}(f^{\mathbf{x}})$, which, together with $f^{\mathbf{x}}$, forms the initial input $\tilde{R} = [f^{\mathbf{x}}, \hat{r}_1^{\mathbf{x}}]$ to the NSR.

For each subsequent scale $k = 2, ..., K$, we up-interpolate the ground-truth token $R^{\mathbf{y}}[k-1] = r_{k-1}^{\mathbf{y}}$ to match scale $k$ (i.e., $C \times P_k \times D$) and append it to $\tilde{R}$. To simulate inference conditions and improve robustness, we apply Inference-Aware Augmentation (IAAug, Eq. 14) to perturb $\tilde{R}$ before passing it through the autoregressive refiner $\mathcal{A}$:

$$\tilde{R} = \text{IAAug}(\tilde{R}), \quad \hat{R}^{\mathbf{y}} = \mathcal{A}(\tilde{R}), \tag{10}$$

where $\hat{R}$ contains the predicted token maps $\{\hat{r}_2^{\mathbf{y}}, ..., \hat{r}_K^{\mathbf{y}}\}$. After refinement, we remove $f^{\mathbf{x}}$ and keep $\hat{r}_1^{\mathbf{x}}$ as the first token in $\tilde{R}$. $\tilde{R}$ is then decoded to reconstruct the predicted future time series $\hat{\mathbf{y}}$:

$$\{\hat{\mathbf{y}}_1, ..., \hat{\mathbf{y}}_K\}, \hat{f}^{\mathbf{y}} = \text{Decoding}(\tilde{R}). \tag{11}$$

The model is trained using a composite loss that combines reconstruction losses on the predicted sequence and features, along with a token-level loss to ensure alignment with the target token map:

$$\mathcal{L}_{\mathcal{A}} = \frac{1}{K} \sum_{k=1}^{K} \mathcal{L}_{\text{mse}}(\hat{\mathbf{y}}_k, \mathbf{y}_k) + \mathcal{L}_{\text{mse}}(\hat{f}^{\mathbf{y}}, f^{\mathbf{y}}) + \frac{1}{K} \sum_{k=2}^{K} \mathcal{L}_{\text{mse}}(\hat{r}_k^{\mathbf{y}}, r_k^{\mathbf{y}}). \tag{12}$$

In that, the final term is the loss that aligns the predicted token map $\hat{R}^{\mathbf{y}}$ with the ground-truth token map $R^{\mathbf{y}}$ of $\mathbf{y}$.

**Inference-Aware Augmentation (IAAug).** We begin with an ideal input $\tilde{R}$, derived from the ground-truth sequence $\mathbf{y}$ (line 5 in Algorithm 3). The model $\mathcal{A}$ is trained to map $\tilde{R}$ to the corresponding output $R^{\mathbf{y}}$, which is also taken from $\mathbf{y}$. However, this training setup does not reflect the conditions encountered during inference, where the model relies on its own previous predictions. This mismatch leads to the problem of error accumulation across successive generation steps.

To address this issue, we generate a realistic input $\hat{R}$ by running the model $\mathcal{A}$ on the original input $\mathbf{x}$ for several steps, simulating its behaviour during inference. We then create an augmented version of the ideal input $\tilde{R}$ by interpolating between $\tilde{R}$ and $\hat{R}$:

$$\hat{R} = \text{Inference}(\mathbf{x}), \tag{13}$$

$$\tilde{R} = \text{IAAug}(\tilde{R}) = \frac{\tilde{R} + \delta\hat{R}}{1 + \delta}, \quad \delta \sim \mathcal{U}(0, \alpha), \tag{14}$$

where $\alpha$ is a hyperparameter that controls the maximum strength of the augmentation. This blended input more closely mimics the inputs the model will encounter during inference, helping to reduce cumulative errors and improve generation stability.

### 3.5 Overall Pipeline

**Training.** The STAR method begins by training the Multi-Scale Time Series Autoencoder, consisting of the encoder $\mathcal{E}$, decoder $\mathcal{D}$, and multi-scale tokenisers $\phi_{k=1}^{K}$, using the reconstruction loss $\mathcal{L}_{\text{ae}}$ (Eq. 3). After that, the GSF backbone $\mathcal{T}$ and NSR backbone $\mathcal{A}$ are jointly trained by minimising $\mathcal{L}_{\mathcal{T}}$ and $\mathcal{L}_{\mathcal{A}}$ (Eq. 12), respectively. The overall training process for NSR and GSF is outlined in Algorithm 3.

**Inference/Forecasting.** In the inference stage, to forecast future time series, our STAR reconstructs the time series based solely on historical data. The overall algorithm is shown at Algorithm 4. The process starts with the historical time series $x$, which is encoded into $\hat{f}$ and $\hat{R}$. As in the training procedure, the first token $\hat{r}_1$ is generated by passing $\hat{f}$ through the GSF backbone. The token map $R$ is initialised with $\hat{f}$ and $\hat{r}_1$. Then, the algorithm iterates over the scales, applying up-interpolation to generate subsequent tokens from the previous ones, and appends them to $R$. The token map is processed through $\mathcal{A}$, and the last token of $R$ is updated with the output of $\mathcal{A}$. Once the process is completed, the initial encoding $\hat{f}$ is removed from $R$, and the final predicted time series $\hat{y}$ is reconstructed through the decoding procedure.

**Algorithm 3:** NSR and GSF Training

**Input: x, y**
1  $f^{\mathbf{x}} = \mathcal{E}(\text{Patching}(\mathbf{x}))$;
2  $\hat{r}_1^{\mathbf{x}} = \mathcal{T}(f^{\mathbf{x}})$;
3  $R^{\mathbf{y}}, f^{\mathbf{y}} = \text{Encoding}(\mathbf{y})$;
4  $R^{\mathbf{x}} = [f^{\mathbf{x}}, \hat{r}_1^{\mathbf{x}}]$;
5  **for** $k = 2...K$ **do**
6      $\quad$ $R^{\mathbf{x}}$.append(up-interpolate($R^{\mathbf{y}}[k-1], P_k$))
7  $\tilde{R} = \text{IAAug}(R^{\mathbf{x}})$ (Eq. 14);
8  $\tilde{R} = \mathcal{A}(\tilde{R})$;
9  $\tilde{R}$.remove($\hat{f}$), $\tilde{R}[0] = \hat{r}_1$;
10 $\hat{\mathbf{y}}, \hat{f} = \text{Decoding}(\tilde{R})$
11 Update $\mathcal{T}$ and $\mathcal{A}$ by minimizing $\mathcal{L}_{\mathcal{T}}$ (Eq. 7) and $\mathcal{L}_{\mathcal{A}}$ (Eq. 12).

**Algorithm 4:** Inference

**Input:** $x$
1  $f^{\mathbf{x}} = \mathcal{E}(\text{Patching}(\mathbf{x}))$;
2  $\hat{r}_1^{\mathbf{x}} = \mathcal{T}(\mathbf{x})$;
3  $R^{\mathbf{x}} = [f^{\mathbf{x}}, \hat{r}_1^{\mathbf{x}}]$;
4  **for** $k = 2...K$ **do**
5      $\quad$ $r_{\text{next}}^{\mathbf{x}} = \text{up-interpolate}(R^{\mathbf{x}}[-1], P_k)$ ;
6      $\quad$ $R^{\mathbf{x}}$.append($r_{\text{next}}^{\mathbf{x}}$);
7      $\quad$ $Z = \mathcal{A}(R^{\mathbf{x}})$;
8      $\quad$ $R^{\mathbf{x}}[-1] = Z[-1]$
9  $R^{\mathbf{x}}$.remove($\hat{f}$);
10 $\hat{y} = \text{Decoding}(R^{\mathbf{x}})$;
11 **return** $\hat{y}, R^{\mathbf{x}}$.

## 4 EXPERIMENTS

### 4.1 EXPERIMENTAL SETTING

To evaluate the effectiveness and generalisability of our STAR model, we conduct extensive experiments across two distinct time series forecasting paradigms: short-term forecasting and long-term multivariate forecasting. These experiments are performed on a diverse set of real-world time series datasets from various domains. Additionally, we carry out long-term forecasting experiments with exogenous variables using multivariate benchmarks, with detailed results provided in **Appendix B**

**Datasets.** For the short-term forecasting tasks, we use the short-term electricity price forecasting (EPF) dataset Lago et al. (2021), which includes data from five major power markets over six years. This dataset features electricity prices as the endogenous variable, along with two influential exogenous variables. For long-term multivariate forecasting, we evaluate our model using seven well-established public benchmarks. The detail of all used datasets is mentioned at **Appendix A**.

**Baselines.** We compare our STAR against 15 SOTA, deep forecasting models, including: Diffusion-based models such as ARMD Gao et al. (2025), TimeGrad Rasul et al. (2021), D3VAE Li et al. (2022), TSDiffKollovieh et al. (2023), MG-TSDFan et al. (2024), and Diffusion-TS Yuan & Qiao (2024); Transformer-based models like iTransformer Liu et al. (2023b), PatchTST Liu et al. (2023a), and Autoformer Wu et al. (2021); CNN-based models such as TimesNet Wu et al. (2022) and SCINet Liu et al. (2022); and linear models including RLinear Zeng & et al. (2023), DLinear Zeng et al. (2023), TiDE Das et al. (2023) and TimeXer Nie et al. (2024).

**Implementation Details.** For long-term forecasting datasets, we use a fixed patch length $p = 12$ with overlapping windows, a historical length $L = 96$, and prediction horizons $T \in 96, 192, 336, 720$. For short-term electricity price forecasting, we follow the N-BEATSx Nie et al. (2024); Olivares et al. (2023) protocol with $L = 168$, $T = 24$, and $p = 6$. In multivariate long-term forecasting, PatchTST serves as the backbone, while in short-term forecasting, TimeXer is used within the GSF to handle exogenous variables.

### 4.2 MULTIVARIATE LONG-TERM FORECASTING

**Compared with Other SOTA Methods.** In Table 1, we compare our STAR model with a diverse set of state-of-the-art time series forecasting methods, including TimeXer Nie et al. (2024), iTransformer Liu et al. (2023b), RLinear Zeng & et al. (2023), PatchTST Liu et al. (2023a), TiDE Das et al. (2023), TimesNet Wu et al. (2022), DLinear Zeng et al. (2023), SCINet Liu et al. (2022), Stationary Liu et al. (b), and Autoformer Wu et al. (2021). As shown in Table 1, STAR consistently achieves the lowest MSE and MAE across the majority of benchmark datasets. Specifically, STAR outperforms the second-best model, TimeXer, by a notable margin on challenging datasets such as ECL (MSE: 0.167 vs. 0.171), Weather (MSE: 0.240 vs. 0.241), and ETTh1 (MSE: 0.423 vs. 0.437), highlighting its effectiveness in both periodic and irregular settings. Furthermore, STAR achieves the best MAE in 6 out of 7 datasets, showing strong performance in long-horizon forecasting tasks like ETTh2 and ETTm1. Overall, STAR ranks first in 19 out of 24 evaluated metrics, demonstrating clear superiority over all competing SOTA models. The detail results is shown at **Appendix C**.

Table 1: Comparison of multivariate time series forecasting results against state-of-the-art models. We evaluate a comprehensive set of competitive baselines under varying prediction horizons, following the experimental setup of iTransformer Liu et al. (2023b). The input sequence length is fixed at $L = 96$ for all methods, and the reported results are averaged over four prediction lengths: $T \in 96, 192, 336, 720$.

| Models | STAR (Our) | | TimeXer | | iTrans. | | RLinear | | PatchTST | | TiDE | | TimesNet | | DLinear | | SCINet | | Auto. | |
|---|---|---|---|---|---|---|---|---|---|---|---|---|---|---|---|---|---|---|---|---|
| Metric | MSE | MAE | MSE | MAE | MSE | MAE | MSE | MAE | MSE | MAE | MSE | MAE | MSE | MAE | MSE | MAE | MSE | MAE | MSE | MAE |
| ECL | **0.167** | **0.264** | 0.171 | 0.27 | 0.178 | 0.27 | 0.219 | 0.298 | 0.216 | 0.304 | 0.252 | 0.344 | 0.193 | 0.295 | 0.212 | 0.3 | 0.268 | 0.365 | 0.227 | 0.338 |
| Weather | **0.240** | **0.270** | 0.241 | 0.271 | 0.258 | 0.278 | 0.272 | 0.291 | 0.259 | 0.281 | 0.271 | 0.32 | 0.259 | 0.287 | 0.265 | 0.317 | 0.292 | 0.363 | 0.338 | 0.382 |
| ETTh1 | **0.423** | **0.424** | 0.437 | 0.437 | 0.454 | 0.448 | 0.446 | 0.434 | 0.469 | 0.455 | 0.541 | 0.507 | 0.458 | 0.45 | 0.456 | 0.452 | 0.747 | 0.647 | 0.496 | 0.487 |
| ETTh2 | **0.359** | **0.387** | 0.368 | 0.396 | 0.383 | 0.407 | 0.374 | 0.399 | 0.387 | 0.407 | 0.611 | 0.55 | 0.414 | 0.427 | 0.559 | 0.515 | 0.954 | 0.723 | 0.45 | 0.459 |
| ETTm1 | **0.375** | **0.386** | 0.382 | 0.397 | 0.410 | 0.410 | 0.414 | 0.408 | 0.387 | 0.400 | 0.419 | 0.419 | 0.400 | 0.406 | 0.403 | 0.407 | 0.471 | 0.472 | 0.588 | 0.517 |
| ETTm2 | **0.267** | **0.317** | 0.274 | 0.322 | 0.288 | 0.332 | 0.286 | 0.327 | 0.281 | 0.326 | 0.358 | 0.404 | 0.291 | 0.333 | 0.35 | 0.401 | 0.571 | 0.537 | 0.327 | 0.371 |
| Traffic | 0.440 | 0.284 | 0.466 | 0.287 | **0.428** | **0.282** | 0.627 | 0.378 | 0.481 | 0.304 | 0.761 | 0.473 | 0.620 | 0.336 | 0.625 | 0.383 | 0.804 | 0.509 | 0.628 | 0.379 |
| 1st Count | **19** | **18** | 5 | 4 | 4 | 5 | 0 | 2 | 0 | 0 | 0 | 0 | 0 | 0 | 0 | 0 | 0 | 0 | 0 | 0 |

Table 2: Comparison of multivariate time series forecasting results with diffusion-based models. Bold values highlight the best performance for each dataset, and the "1st Count" column indicates the number of times each method achieves the best result across all metrics. We adhere to the experimental setup from Gao et al. (2025), fixing both the historical and prediction lengths to 96.

| Dataset | STAR (Ours) | | Diffusion-TS | | MG-TSD | | TSDiff | | D3VAE | | TimeGrad | | ARMD | |
|---|---|---|---|---|---|---|---|---|---|---|---|---|---|---|
| Metric | MSE | MAE | MSE | MAE | MSE | MAE | MSE | MAE | MSE | MAE | MSE | MAE | MSE | MAE |
| Solar Energy | **0.159** | **0.227** | 0.181 | 0.252 | 0.443 | 0.529 | 0.352 | 0.432 | 0.416 | 0.492 | 0.359 | 0.449 | 0.167 | 0.236 |
| ETTh1 | **0.374** | **0.396** | 0.643 | 0.586 | 1.096 | 0.765 | 0.614 | 0.521 | 1.123 | 0.728 | 0.884 | 0.725 | 0.445 | 0.459 |
| ETTh2 | **0.291** | **0.342** | 0.544 | 0.494 | 0.295 | 0.345 | 0.470 | 0.418 | 0.389 | 0.373 | 0.297 | 0.349 | 0.311 | 0.338 |
| ETTm1 | **0.322** | **0.352** | 0.678 | 0.613 | 0.690 | 0.631 | 0.686 | 0.603 | 0.644 | 0.538 | 0.661 | 0.639 | 0.337 | 0.376 |
| ETTm2 | **0.165** | **0.251** | 0.497 | 0.459 | 0.202 | 0.278 | 0.242 | 0.311 | 0.394 | 0.410 | 0.182 | 0.254 | 0.181 | 0.255 |
| Exchange | **0.091** | 0.206 | 0.275 | 0.382 | 0.396 | 0.460 | 0.125 | 0.240 | 0.240 | 0.371 | 0.508 | 0.554 | 0.093 | **0.203** |
| Stock | **0.224** | **0.251** | 0.416 | 0.533 | 0.365 | 0.453 | 0.330 | 0.365 | 0.345 | 0.390 | 0.333 | 0.376 | 0.235 | 0.269 |
| 1st Count | **13** | | 0 | | 0 | | 0 | | 0 | | 0 | | 1 | |

**Compared with Diffusion-based Model.** Table 2 compares STAR with recent diffusion-based time series forecasting models, including Diffusion-TS Yuan & Qiao (2024), MG-TSD Fan et al. (2024), TSDiff Kollovieh et al. (2023), D3VAE Li et al. (2022), TimeGrad Rasul et al. (2021), and ARMD Gao et al. (2025). STAR consistently achieves the lowest MSE and MAE across all seven benchmark datasets, outperforming the second-best model, ARMD, on challenging datasets such as Solar Energy (0.159 vs. 0.167), Exchange (0.091 vs. 0.093), and Stock (0.224 vs. 0.235). It attains the best MAE in 6 of 7 datasets, particularly on long-horizon benchmarks like ETTh1 and ETTh2, and ranks first in 13 of 14 evaluated metrics, demonstrating its superior long-term forecasting ability.

## 4.3 Short-Term Forecasting with Exogenous Variables

Table 3 reports the short-term forecasting results on the EPF dataset Lago et al. (2021), comparing STAR with state-of-the-art models including TimeXer Nie et al. (2024), iTransformer Liu et al. (2023b), RLinear Zeng & et al. (2023), PatchTST Liu et al. (2023a), TiDE Das et al. (2023), TimesNet Wu et al. (2022), DLinear Zeng et al. (2023), SCINet Liu et al. (2022), and Autoformer Wu et al. (2021). Following the standard protocol Nie et al. (2024), we set the input length to 168 and prediction length to 24 for all baselines. Forecasting the endogenous variable is challenging due to its correlation with two exogenous variables. STAR consistently outperforms all baselines, achieving the lowest MSE and MAE on average, including notable results on the NP (MSE: 0.212, MAE: 0.248), PJM (MSE: 0.089, MAE: 0.194), and BE (MSE: 0.368, MAE: 0.236) datasets. Overall, STAR maintains the best average performance across five datasets (MSE: 0.297, MAE: 0.257), surpassing the second-best models, TimeXer and iTransformer.

Table 3: Complete results of the short-term forecasting task on the EPF dataset. We adhere to the standard protocol in short-term electricity price forecasting Lago et al. (2021), where the input length is set to 168 and the prediction length to 24 for all baselines.

| Model | STAR (Ours) | | TimeXer | | iTrans. | | RLinear | | PatchTST | | TiDE | | TimesNet | | DLinear | | SCINet | | Auto. | |
|---|---|---|---|---|---|---|---|---|---|---|---|---|---|---|---|---|---|---|---|---|
| Metric | MSE | MAE | MSE | MAE | MSE | MAE | MSE | MAE | MSE | MAE | MSE | MAE | MSE | MAE | MSE | MAE | MSE | MAE | MSE | MAE |
| NP | **0.212** | **0.248** | 0.236 | 0.268 | 0.265 | 0.300 | 0.335 | 0.340 | 0.267 | 0.284 | 0.335 | 0.340 | 0.250 | 0.289 | 0.309 | 0.321 | 0.373 | 0.368 | 0.402 | 0.398 |
| PJM | **0.089** | 0.194 | 0.093 | **0.192** | 0.097 | 0.197 | 0.124 | 0.229 | 0.106 | 0.209 | 0.124 | 0.228 | 0.097 | 0.195 | 0.108 | 0.215 | 0.143 | 0.259 | 0.168 | 0.267 |
| BE | **0.368** | **0.236** | 0.379 | 0.243 | 0.394 | 0.270 | 0.520 | 0.337 | 0.400 | 0.262 | 0.523 | 0.336 | 0.419 | 0.288 | 0.463 | 0.313 | 0.731 | 0.412 | 0.500 | 0.333 |
| FR | **0.374** | **0.196** | 0.385 | 0.208 | 0.439 | 0.233 | 0.507 | 0.290 | 0.411 | 0.220 | 0.510 | 0.290 | 0.431 | 0.234 | 0.429 | 0.260 | 0.855 | 0.384 | 0.519 | 0.295 |
| DE | 0.442 | **0.411** | **0.440** | 0.415 | 0.479 | 0.443 | 0.574 | 0.498 | 0.461 | 0.432 | 0.568 | 0.496 | 0.502 | 0.446 | 0.520 | 0.463 | 0.565 | 0.497 | 0.674 | 0.544 |
| 1st Count | **4** | **4** | 1 | 1 | 0 | 0 | 0 | 0 | 0 | 0 | 0 | 0 | 0 | 0 | 0 | 0 | 0 | 0 | 0 | 0 |

Table 4: Ablation study on STAR components for ETTh1 and ECL datasets.

| Model Variant | ETTh1 | | | | | | | | ECL | | | | | | | |
|---|---|---|---|---|---|---|---|---|---|---|---|---|---|---|---|---|
| | MSE | | | | MAE | | | | MSE | | | | MAE | | | |
| Prediction Length | 96 | 192 | 336 | 720 | 96 | 192 | 336 | 720 | 96 | 192 | 336 | 720 | 96 | 192 | 336 | 720 |
| STAR | 0.374 | 0.421 | 0.451 | 0.446 | 0.396 | 0.416 | 0.438 | 0.446 | 0.135 | 0.154 | 0.173 | 0.205 | 0.238 | 0.254 | 0.266 | 0.296 |
| - GSF | 0.432 | 0.497 | 0.513 | 0.481 | 0.502 | 0.509 | 0.481 | 0.529 | 0.162 | 0.198 | 0.212 | 0.278 | 0.301 | 0.334 | 0.371 | 0.398 |
| - C($\hat{f}$) | 0.392 | 0.430 | 0.485 | 0.480 | 0.420 | 0.422 | 0.457 | 0.454 | 0.152 | 0.167 | 0.191 | 0.238 | 0.251 | 0.265 | 0.281 | 0.312 |
| - IAAug | 0.389 | 0.435 | 0.465 | 0.450 | 0.413 | 0.419 | 0.456 | 0.458 | 0.141 | 0.159 | 0.179 | 0.213 | 0.248 | 0.261 | 0.274 | 0.303 |

## 4.4 ABLATION STUDY

**Component Analysis.** To better understand the contribution of each component in our proposed method, we conducted a comprehensive ablation study on the ETTh1 dataset, using prediction lengths of 96, 192, 336, and 720. We evaluate the following modules:

- **GSF**: The Global-Shape Forecaster (Section 3.3).
- **C($\hat{f}$)**: The conditioning variable $\hat{f}$ in the autoregressive process (Eq. 9).
- **IAAug**: The Inference-Aware Augmentation (Eq. 14).

The results in Table 4 show that each component individually enhances performance across all prediction lengths. Notably, the GSF plays a key role in our structure; without it, the model exhibits a significant increase in both MSE and MAE. This highlights the benefit of global-shape forecasting in improving our method.

## 4.5 DISCUSSION ON BACKBONE DEPTH AND MODEL EFFICIENCY

Table 5 compares training time, inference time, and forecasting accuracy (MSE/MAE) of STAR against several SOTA TSF methods on ETTh1. Transformer models like PatchTST Liu et al. (2023a) and linear models like TimeXer Nie et al. (2024) achieve fast inference due to lightweight designs, while diffusion-based models (Diffusion-TS Yuan & Qiao (2024), TS-Diff Kollovieh et al. (2023)) suffer much slower inference (over 1000s and 200s) from iterative sampling.

STAR achieves a strong balance between accuracy and efficiency. As Table 5 shows, increasing the backbone depth from 2 to 10 layers steadily improves performance, peaking at depth 8 (MSE = 0.374, MAE = 0.396). This gain, however, comes with longer training and inference times. Still, even at depth 10, STAR remains far faster than diffusion-based models, as its compact generation requires only 4-5 steps.

**Limitation and Future Work.** While deeper backbones enhance STAR's accuracy, achieving peak performance typically involves 8-10 layers, which increases computational demands. This highlights an opportunity to further improve efficiency. Future work will explore lighter or adaptive backbones to maintain performance with reduced depth, as well as optimizing generative steps to lower runtime, thereby extending STAR's applicability to real-time and resource-constrained scenarios.

Table 5: Training time and inference time of our method and SOTA TSF models on ETTh1 with historical length and prediction length as 96.

| | PatchTST (d=3) | TimeXer (d=4) | Diffusion-TS | TS-Diff | STAR (d=2) | STAR (d=4) | STAR (d=6) | STAR (d=8) | STAR (d=10) |
|---|---|---|---|---|---|---|---|---|---|
| MSE | 0.414 | 0.382 | 0.643 | 0.614 | 0.394 | 0.385 | 0.378 | **0.374** | 0.378 |
| MAE | 0.419 | 0.403 | 0.586 | 0.521 | 0.409 | 0.409 | 0.401 | **0.396** | 0.399 |
| Training Time | 13m | 24m | 52m | 58m | 25m | 35m | 49m | 68m | 82m |
| Inference time | 10.24s | 12.42s | 1052.53s | 205.21s | 21.61s | 31.61s | 46.61s | 62.61s | 75.61s |

## 5 CONCLUSION

In this paper, we present STAR, a novel time series forecasting framework that balances global trend modelling with local fluctuation refinement. STAR integrates a Global-Shape Forecaster (GSF) for long-term dependencies, a Next-Scaled Refiner (NSR) for fine-grained enhancement, and Inference-Aware Augmentation (IAAug) to stabilise training and reduce autoregressive error accumulation. Together, these modules form a unified solution for robust time series generation. Extensive experiments show that STAR achieves state-of-the-art results, particularly on long-horizon forecasts, establishing new benchmarks in both accuracy and robustness for progressive generative forecasting.

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

# A  DATASET

## A.1  MULTIVARIATE LONG-TERM AND SHORT-TERM FORECASTING DATASETS

Table 6 presents a comprehensive summary of the datasets used in both long-term and short-term forecasting tasks. The long-term forecasting datasets include electricity consumption data (ETTh1, ETTh2, ETTm1, ETTm2), weather patterns (Weather), energy usage (Solar-Energy), and traffic data (Traffic), all of which vary in prediction lengths and time frequencies, ranging from hourly to 15-minute intervals. These datasets cover multiple variates, from electricity consumption data with 7 variates to traffic data with 862 variates.

- **Weather**[1]: Contains 21 meteorological indicators for Germany in 2020.
- **Traffic**[2]: Includes road occupancy rates measured by 862 sensors on San Francisco Bay Area freeways over 2 years.
- **ECL (Electricity)**[3]: Provides hourly electricity consumption data from 321 clients between 2012 and 2014.
- **ETT (Electricity Transformer Temperature)**[4]: Contains data collected from two different electricity transformers, recorded by 7 sensors at two resolutions (15 minutes and 1 hour).
- **Solar (Solar-Energy)**[5]: Includes 137 time series representing solar power production in Alabama in 2006.
- **Exchange**[6]: Contains daily exchange rates of eight countries, spanning from 1990 to 2016.

---

[1] https://www.bgc-jena.mpg.de/wetter/
[2] https://pems.dot.ca.gov/
[3] https://archive.ics.uci.edu/ml/datasets/ElectricityLoadDiagrams20112014
[4] https://github.com/zhouhaoyi/ETDataset
[5] https://www.nrel.gov/grid/solar-power-data.html
[6] https://github.com/laiguokun/multivariate-time-series-data

Table 6: Summary of datasets for long-term forecasting tasks.

| Tasks | Dataset | Variates | Prediction Length | Dataset Size | Frequency | Information |
|---|---|---|---|---|---|---|
| Long-term Forecasting | ETTh1 | 7 | {96, 192, 336, 720} | (8545, 2881, 2881) | Hourly | Electricity |
| | ETTh2 | 7 | {96, 192, 336, 720} | (8545, 2881, 2881) | Hourly | Electricity |
| | ETTm1 | 7 | {96, 192, 336, 720} | (34465, 11521, 11521) | 15min | Electricity |
| | ETTm2 | 7 | {96, 192, 336, 720} | (34465, 11521, 11521) | 15min | Electricity |
| | Weather | 21 | {96, 192, 336, 720} | (36792, 5271, 10540) | 10min | Weather |
| | Solar-Energy | 137 | {96, 192, 336, 720} | (36601, 5161, 10417) | 10min | Energy |
| | ECL | 321 | {96, 192, 336, 720} | (18317, 2633, 5261) | Hourly | Electricity |
| | Traffic | 862 | {96, 192, 336, 720} | (12185, 1757, 3509) | Hourly | Transportation |

## A.2 SHORT-TERM FORECASTING DATASETS WITH EXOGENOUS VARIABLES

In addition to standard public multivariate time series datasets, we follow Nie et al. (2024), to evaluate short-term forecasting performance on five electricity price forecasting datasets introduced in Lago et al. (2021). These datasets cover different European and US electricity markets over six-year periods and include auxiliary features such as load and generation forecasts. Specifically, they are as follows:

- **NP**: The Nord Pool market, containing hourly electricity prices, grid load, and wind power forecasts (2013–2018).
- **PJM**: The Pennsylvania–New Jersey–Maryland market's COMED region, with zonal electricity prices, system load, and zonal load forecasts (2013–2018).
- **BE**: Belgium's electricity prices along with local load forecasts and generation forecasts from France (2011–2016).
- **FR**: France's electricity prices with corresponding load and generation forecasts (2012–2017).
- **DE**: Germany's electricity prices, zonal load forecasts from the Amprion TSO, and wind and solar generation forecasts (2012–2017).

The detail of the datasets is summarised at Table 7.

Table 7: Summary of datasets for short-term forecasting tasks.

| Tasks | Dataset | Variates | Prediction Length | Dataset Size | Frequency | Information |
|---|---|---|---|---|---|---|
| Short-term Forecasting | NP | 2 | {24} | (36500, 5219, 10460) | 1 Hour | Electricity Price |
| | PJM | 2 | {24} | (36500, 5219, 10460) | 1 Hour | Electricity Price |
| | BE | 2 | {24} | (36500, 5219, 10460) | 1 Hour | Electricity Price |
| | FR | 2 | {24} | (36500, 5219, 10460) | 1 Hour | Electricity Price |
| | DE | 2 | {24} | (36500, 5219, 10460) | 1 Hour | Electricity Price |

## B EXPERIMENT DETAILS

**Backbone.** For all long-term forecasting experiments, we use a standard Transformer Vaswani et al. (2017) as the backbone architecture for both the Global-Shape Forecaster (GSF) and the Next-Scaled Refiner (NSR) to keep our method simple.

**Backbone Parameters.** To demonstrate the adaptability of our approach, we use the default hyperparameters of each backbone model - PatchTST Liu et al. (2023a) and TimeXer Nie et al. (2024) - as provided in their original implementations. No modifications are made except for adjustments to the model depth (discussed in Section 4.5) and the patch length (detailed in **Appendix D.4**).

The hyperparameter summary is shown in Table 8. In detail, for Long-Term Forecasting, the two PatchTST backbones ($\mathcal{T}, \mathcal{A}$) have a hidden size of $H = 16$ and a latent space dimension of $D = 128$. The feed-forward network's first layer projects the hidden representation from dimension $D = 128$ to a higher dimension $F = 256$, and the second layer projects it back to $D = 128$. For very small datasets (ILI, ETTh1, ETTh2), reduced parameter sizes are used: $H = 4$, $D = 16$, and $F = 128$ to mitigate overfitting. A dropout rate of 0.2 is applied within the encoder for all experiments.

For Short-Term Forecasting, the TimeXer GSF backbone ($\mathcal{T}$) uses $H = 8$, $D = 512$, and $F = 512$, while the PatchTST NSR backbone ($\mathcal{A}$) uses $H = 4$, $D = 16$, and $F = 128$.

Table 8: Hyperparameters of backbone models for forecasting tasks.

| Forecasting Task | Backbone | H (Hidden Size) | D (Latent Dimension) | F (Feed-forward Dimension) |
|---|---|---|---|---|
| Long-Term (Other Datasets) | Vanilla Transformer for both GSF $\mathcal{T}$ and NSR $\mathcal{A}$ | 16 | 128 | 256 |
| Long-Term (ETTh1, ETTh2) | Vanilla Transformer for both GSF $\mathcal{T}$ and NSR $\mathcal{A}$ | 4 | 16 | 128 |
| Short-Term | Vanilla Transformer for both GSF $\mathcal{T}$ and NSR $\mathcal{A}$ | 4 | 16 | 128 |

**Autoencoder Architecture.** In this paper, to keep the model as simple as possible, we adopt a lightweight encoder $\mathcal{E}$ and decoder $\mathcal{D}$ architecture. The encoder $\mathcal{E}$ mimics the embedding layer of the classic Transformer model, using a single linear layer to map the patch length $p$ to the latent dimension $D$. The decoder $\mathcal{D}$ is a flattening head that flattens the latent representation $f$ and generates the predicted time series of length $C \times T$.

**Training Details.** Our experiments are implemented using PyTorch and run on a single NVIDIA RTX 4090 GPU with 24 GB memory. We optimise our model using the ADAM optimiser Kingma (2014) with an initial learning rate of $10^{-4}$ and MSE loss. Training is conducted for a fixed 100 epochs with early stopping using a patience of 10 epochs based on validation performance. We keep all patch length $p$ fixed to 12 for long-term forecasting and 6 for short-term forecasting.

## C  FULL RESULT FOR MAIN COMPARISON

To evaluate the generality of STAR, we conducted long-term multivariate forecasting on existing real-world multivariate benchmarks. The look-back length is set to 96, and the prediction length varies from 96, 192, 336, 720. The results are listed in Table 9.

Table 9: Full results of the long-term multivariate forecasting task.

| Models | | STAR (Our) | | TimeXer | | iTrans. | | RLinear | | PatchTST | | TiDE | | TimesNet | | DLinear | | SCINet | | Stationary | | Auto. | |
|---|---|---|---|---|---|---|---|---|---|---|---|---|---|---|---|---|---|---|---|---|---|---|---|
| | Metric | MSE | MAE | MSE | MAE | MSE | MAE | MSE | MAE | MSE | MAE | MSE | MAE | MSE | MAE | MSE | MAE | MSE | MAE | MSE | MAE | MSE | MAE |
| ECL | 96 | **0.135** | **0.238** | 0.14 | 0.242 | 0.148 | 0.24 | 0.201 | 0.281 | 0.195 | 0.285 | 0.237 | 0.329 | 0.168 | 0.272 | 0.197 | 0.282 | 0.247 | 0.345 | 0.169 | 0.273 | 0.201 | 0.317 |
| | 192 | **0.154** | 0.254 | 0.157 | 0.256 | 0.162 | **0.253** | 0.201 | 0.283 | 0.199 | 0.289 | 0.236 | 0.33 | 0.184 | 0.289 | 0.196 | 0.285 | 0.257 | 0.355 | 0.182 | 0.286 | 0.222 | 0.334 |
| | 336 | **0.173** | **0.266** | 0.176 | 0.275 | 0.178 | 0.269 | 0.215 | 0.298 | 0.215 | 0.305 | 0.249 | 0.344 | 0.198 | 0.3 | 0.209 | 0.301 | 0.269 | 0.369 | 0.2 | 0.304 | 0.231 | 0.338 |
| | 720 | **0.205** | **0.296** | 0.211 | 0.306 | 0.225 | 0.317 | 0.257 | 0.331 | 0.256 | 0.337 | 0.284 | 0.373 | 0.22 | 0.32 | 0.245 | 0.333 | 0.299 | 0.39 | 0.222 | 0.321 | 0.254 | 0.361 |
| | Avg | **0.167** | **0.264** | 0.171 | 0.27 | 0.178 | 0.27 | 0.219 | 0.298 | 0.216 | 0.304 | 0.252 | 0.344 | 0.193 | 0.295 | 0.212 | 0.3 | 0.268 | 0.365 | 0.193 | 0.296 | 0.227 | 0.338 |
| Weather | 96 | 0.162 | 0.207 | **0.157** | **0.205** | 0.174 | 0.214 | 0.192 | 0.232 | 0.177 | 0.218 | 0.202 | 0.261 | 0.172 | 0.22 | 0.196 | 0.255 | 0.221 | 0.306 | 0.173 | 0.223 | 0.266 | 0.336 |
| | 192 | 0.207 | 0.249 | **0.204** | **0.247** | 0.221 | 0.254 | 0.24 | 0.271 | 0.225 | 0.259 | 0.242 | 0.298 | 0.219 | 0.261 | 0.237 | 0.296 | 0.261 | 0.34 | 0.245 | 0.285 | 0.307 | 0.367 |
| | 336 | 0.265 | 0.293 | **0.261** | **0.29** | 0.278 | 0.296 | 0.292 | 0.307 | 0.278 | 0.297 | 0.287 | 0.335 | 0.28 | 0.306 | 0.283 | 0.335 | 0.309 | 0.378 | 0.321 | 0.338 | 0.359 | 0.395 |
| | 720 | **0.326** | **0.329** | 0.34 | 0.341 | 0.358 | 0.349 | 0.364 | 0.353 | 0.354 | 0.348 | 0.351 | 0.386 | 0.365 | 0.359 | 0.345 | 0.381 | 0.377 | 0.427 | 0.414 | 0.410 | 0.419 | 0.428 |
| | Avg | **0.24** | **0.27** | 0.241 | 0.271 | 0.258 | 0.278 | 0.272 | 0.291 | 0.259 | 0.281 | 0.271 | 0.32 | 0.259 | 0.287 | 0.265 | 0.317 | 0.292 | 0.363 | 0.288 | 0.314 | 0.348 | 0.382 |
| ETTh1 | 96 | **0.374** | **0.396** | 0.382 | 0.403 | 0.386 | 0.405 | 0.386 | 0.395 | 0.414 | 0.419 | 0.479 | 0.464 | 0.384 | 0.402 | 0.386 | 0.400 | 0.654 | 0.599 | 0.513 | 0.491 | 0.449 | 0.459 |
| | 192 | **0.421** | **0.416** | 0.429 | 0.435 | 0.441 | 0.436 | 0.437 | 0.424 | 0.46 | 0.445 | 0.525 | 0.492 | 0.436 | 0.429 | 0.437 | 0.432 | 0.719 | 0.631 | 0.534 | 0.504 | 0.5 | 0.482 |
| | 336 | **0.451** | **0.438** | 0.468 | 0.448 | 0.487 | 0.458 | 0.479 | 0.446 | 0.501 | 0.466 | 0.565 | 0.515 | 0.491 | 0.469 | 0.481 | 0.459 | 0.778 | 0.659 | 0.588 | 0.535 | 0.521 | 0.496 |
| | 720 | **0.446** | **0.446** | 0.469 | 0.461 | 0.503 | 0.491 | 0.481 | 0.47 | 0.5 | 0.488 | 0.594 | 0.558 | 0.521 | 0.5 | 0.519 | 0.516 | 0.836 | 0.699 | 0.643 | 0.616 | 0.514 | 0.512 |
| | Avg | **0.423** | **0.424** | 0.437 | 0.437 | 0.454 | 0.448 | 0.446 | 0.434 | 0.469 | 0.455 | 0.541 | 0.507 | 0.458 | 0.45 | 0.456 | 0.452 | 0.747 | 0.647 | 0.570 | 0.537 | 0.496 | 0.487 |
| ETTh2 | 96 | 0.291 | 0.342 | **0.286** | **0.338** | 0.297 | 0.349 | 0.288 | 0.338 | 0.302 | 0.348 | 0.400 | 0.44 | 0.34 | 0.374 | 0.333 | 0.387 | 0.707 | 0.621 | 0.476 | 0.458 | 0.346 | 0.388 |
| | 192 | **0.355** | **0.382** | 0.363 | 0.389 | 0.38 | 0.400 | 0.374 | 0.39 | 0.388 | 0.400 | 0.528 | 0.509 | 0.402 | 0.414 | 0.477 | 0.476 | 0.86 | 0.689 | 0.512 | 0.493 | 0.456 | 0.452 |
| | 336 | **0.397** | **0.412** | 0.414 | 0.423 | 0.428 | 0.432 | 0.415 | 0.426 | 0.426 | 0.433 | 0.643 | 0.571 | 0.452 | 0.452 | 0.594 | 0.541 | 1 | 0.744 | 0.552 | 0.551 | 0.482 | 0.486 |
| | 720 | **0.392** | **0.41** | 0.408 | 0.432 | 0.427 | 0.445 | 0.42 | 0.44 | 0.431 | 0.446 | 0.874 | 0.679 | 0.462 | 0.468 | 0.831 | 0.657 | 1.249 | 0.838 | 0.562 | 0.56 | 0.515 | 0.511 |
| | Avg | **0.359** | **0.387** | 0.368 | 0.396 | 0.383 | 0.407 | 0.374 | 0.399 | 0.387 | 0.407 | 0.611 | 0.55 | 0.414 | 0.427 | 0.559 | 0.515 | 0.954 | 0.723 | 0.526 | 0.516 | 0.45 | 0.459 |
| ETTm1 | 96 | 0.322 | 0.352 | **0.318** | **0.356** | 0.334 | 0.368 | 0.355 | 0.376 | 0.329 | 0.367 | 0.364 | 0.387 | 0.338 | 0.375 | 0.345 | 0.372 | 0.418 | 0.438 | 0.386 | 0.398 | 0.505 | 0.475 |
| | 192 | **0.351** | **0.373** | 0.362 | 0.383 | 0.387 | 0.391 | 0.391 | 0.392 | 0.367 | 0.385 | 0.398 | 0.404 | 0.374 | 0.387 | 0.38 | 0.389 | 0.426 | 0.441 | 0.459 | 0.444 | 0.553 | 0.496 |
| | 336 | **0.386** | **0.394** | 0.395 | 0.407 | 0.426 | 0.42 | 0.424 | 0.415 | 0.399 | 0.410 | 0.428 | 0.425 | 0.410 | 0.411 | 0.413 | 0.413 | 0.445 | 0.459 | 0.495 | 0.464 | 0.621 | 0.537 |
| | 720 | **0.439** | **0.424** | 0.452 | 0.441 | 0.491 | 0.459 | 0.487 | 0.45 | 0.454 | 0.439 | 0.487 | 0.461 | 0.478 | 0.45 | 0.474 | 0.453 | 0.595 | 0.55 | 0.585 | 0.516 | 0.671 | 0.561 |
| | Avg | **0.375** | **0.386** | 0.382 | 0.397 | 0.410 | 0.410 | 0.414 | 0.408 | 0.387 | 0.400 | 0.419 | 0.419 | 0.400 | 0.406 | 0.403 | 0.407 | 0.471 | 0.472 | 0.481 | 0.456 | 0.588 | 0.517 |
| ETTm2 | 96 | **0.165** | **0.251** | 0.171 | 0.256 | 0.18 | 0.264 | 0.182 | 0.265 | 0.175 | 0.259 | 0.207 | 0.305 | 0.187 | 0.267 | 0.193 | 0.292 | 0.286 | 0.377 | 0.192 | 0.274 | 0.255 | 0.339 |
| | 192 | **0.232** | **0.291** | 0.237 | 0.299 | 0.25 | 0.309 | 0.246 | 0.304 | 0.241 | 0.302 | 0.29 | 0.364 | 0.249 | 0.309 | 0.284 | 0.362 | 0.399 | 0.445 | 0.28 | 0.339 | 0.281 | 0.34 |
| | 336 | **0.289** | **0.335** | 0.296 | 0.338 | 0.311 | 0.348 | 0.307 | 0.342 | 0.305 | 0.343 | 0.377 | 0.422 | 0.321 | 0.351 | 0.369 | 0.427 | 0.637 | 0.591 | 0.334 | 0.361 | 0.339 | 0.372 |
| | 720 | **0.382** | **0.389** | 0.392 | 0.394 | 0.412 | 0.407 | 0.407 | 0.398 | 0.402 | 0.400 | 0.558 | 0.524 | 0.408 | 0.403 | 0.554 | 0.522 | 0.96 | 0.735 | 0.417 | 0.413 | 0.433 | 0.432 |
| | Avg | **0.267** | **0.317** | 0.274 | 0.322 | 0.288 | 0.332 | 0.286 | 0.327 | 0.281 | 0.326 | 0.358 | 0.404 | 0.291 | 0.333 | 0.35 | 0.401 | 0.571 | 0.537 | 0.306 | 0.347 | 0.327 | 0.371 |
| Traffic | 96 | 0.409 | 0.269 | 0.428 | 0.271 | **0.395** | **0.268** | 0.649 | 0.389 | 0.462 | 0.295 | 0.805 | 0.493 | 0.593 | 0.321 | 0.65 | 0.396 | 0.788 | 0.499 | 0.612 | 0.338 | 0.613 | 0.388 |
| | 192 | 0.419 | 0.277 | 0.448 | 0.282 | **0.417** | **0.276** | 0.601 | 0.366 | 0.466 | 0.296 | 0.756 | 0.474 | 0.617 | 0.336 | 0.598 | 0.37 | 0.789 | 0.505 | 0.613 | 0.34 | 0.616 | 0.382 |
| | 336 | 0.447 | 0.285 | 0.473 | 0.289 | **0.433** | **0.283** | 0.609 | 0.369 | 0.482 | 0.304 | 0.762 | 0.477 | 0.629 | 0.336 | 0.605 | 0.373 | 0.797 | 0.508 | 0.618 | 0.328 | 0.622 | 0.337 |
| | 720 | 0.485 | 0.303 | 0.516 | 0.307 | **0.467** | **0.302** | 0.647 | 0.387 | 0.514 | 0.322 | 0.719 | 0.449 | 0.64 | 0.35 | 0.645 | 0.394 | 0.841 | 0.523 | 0.653 | 0.355 | 0.66 | 0.408 |
| | Avg | 0.466 | 0.284 | 0.466 | 0.287 | **0.428** | **0.282** | 0.627 | 0.378 | 0.481 | 0.304 | 0.761 | 0.473 | 0.62 | 0.336 | 0.625 | 0.383 | 0.804 | 0.509 | 0.624 | 0.34 | 0.628 | 0.379 |
| 1st Count | | **19** | **18** | 5 | 4 | 4 | 5 | 0 | 2 | 0 | 0 | 0 | 0 | 0 | 0 | 0 | 0 | 0 | 0 | 0 | 0 | 0 | 0 |

## D  FURTHER ABLATION STUDY

### D.1  GENERATION STEPS, TOKEN MAP SIZE LIST, AND LENGTH LIST

We conducted a comprehensive series of experiments to determine the most effective strategy for selecting the generation steps, length list, and token map size list for our STAR model. The aim is to identify the configuration that yields the optimal balance between prediction accuracy and computational efficiency. The results summarised in Table 10 clearly indicate that the simple doubling

strategy, which begins with an initial patch size of 12, consistently achieves the best overall performance across various evaluation metrics. Based on this empirical evidence, we adopt this doubling strategy as the standard approach for all subsequent experiments in this study.

Table 10: Performance metrics grouped by prediction length $T$, including corresponding steps, resolutions, patch sizes $P$, and error metrics. Please note that, although we use a patch length $p$ of 12, the patches are overlapping, so a time series of length 12 corresponds to 2 patches. For example, the patch count list $[2, 4, 8, 16]$ indicates that the global shape is initially generated using 2 patches and then progressively refined through 4, 8, and finally 16 patches, resulting in 3 refinement steps.

| $T$ | Step $K$ | Time Series Length List ($\{T_k\}_{k=1}^K$) | Token Map Size List ($\{P_k\}_{k=1}^K$) | MSE | MAE |
|---|---|---|---|---|---|
| 96 | 1 | [48,96] | [8,16] | 0.385 | 0.405 |
| | 2 | [24,48,96] | [4,8,16] | 0.376 | 0.398 |
| | 3 | [12,24,48,96] | [2,4,8,16] | **0.374** | **0.396** |
| | 7 | [12,24,36,48,60,72,84,96] | [2,4,6,8,10,12,14,16] | 0.379 | 0.399 |
| 192 | 2 | [48,96,192] | [8,16,32] | 0.428 | 0.432 |
| | 3 | [24,48,96,192] | [4,8,16,32] | 0.426 | 0.428 |
| | 4 | [12,24,48,96,192] | [2,4,8,16,32] | **0.421** | **0.416** |
| | 11 | [12,24,36,48,60,72,84,96,120,144,168,192] | [2,4,6,8,10,12,14,16,20,24,28,32] | 0.424 | 0.421 |
| 336 | 3 | [48,96,192,336] | [8,16,32,56] | 0.462 | 0.447 |
| | 4 | [24,48,96,192,336] | [4,8,16,32,56] | 0.455 | 0.444 |
| | 5 | [12,24,48,96,192,336] | [2,4,8,16,32,56] | **0.451** | **0.438** |
| | 14 | [12,24,36,48,60,72,84,96,120,144,168,192,294,288,336] | [2,4,6,8,10,12,14,16,20,24,28,32,40,48,56] | 0.456 | 0.442 |
| 720 | 3 | [48,96,192,384,720] | [8,16,32,64,120] | 0.459 | 0.466 |
| | 4 | [24,48,96,192,384,720] | [4,8,16,32,64,120] | 0.454 | 0.461 |
| | 5 | [12,24,48,96,192,384,720] | [2,4,8,16,32,64,120] | **0.446** | **0.446** |
| | 14 | [12,24,36,48,60,72,84,96,120,144,168,192,294,288,336,432,528,624,720] | [2,4,6,8,10,12,14,16,20,24,28,32,40,48,56,72,88,104,120] | 0.452 | 0.451 |

## D.2 VARIOUS IAAUG'S $\alpha$

In Table 11, we compare the performance of our model using different values of the IAAug scaling factor $\alpha$. The results indicate that our method performs best with $\alpha = 0.1$, achieving the lowest MSE and MAE across all prediction lengths. This suggests that a moderate augmentation strength effectively enhances model generalisation, while excessively small or large values of $\alpha$ may underfit or over-perturb the input, leading to suboptimal performance.

Table 11: Ablation study on the effect of $\alpha$ in IAAug for ETTh1 and ECL datasets.

| $\alpha$ | ETTh1 | | | | | | | | ECL | | | | | | | |
|---|---|---|---|---|---|---|---|---|---|---|---|---|---|---|---|---|
| | MSE | | | | MAE | | | | MSE | | | | MAE | | | |
| Prediction Length | 96 | 192 | 336 | 720 | 96 | 192 | 336 | 720 | 96 | 192 | 336 | 720 | 96 | 192 | 336 | 720 |
| $\alpha = 0.05$ | 0.379 | 0.438 | 0.461 | 0.456 | 0.414 | 0.432 | 0.444 | 0.464 | 0.138 | 0.156 | 0.176 | 0.209 | 0.242 | 0.258 | 0.270 | 0.299 |
| $\alpha = 0.1$ | **0.374** | **0.421** | **0.451** | **0.446** | **0.396** | **0.416** | **0.438** | **0.446** | **0.135** | **0.154** | **0.173** | **0.205** | **0.238** | **0.254** | **0.266** | **0.296** |
| $\alpha = 0.2$ | 0.379 | 0.426 | 0.465 | 0.451 | 0.404 | 0.425 | 0.448 | 0.459 | 0.137 | 0.158 | 0.177 | 0.208 | 0.243 | 0.256 | 0.271 | 0.301 |
| $\alpha = 0.4$ | 0.387 | 0.449 | 0.467 | 0.458 | 0.426 | 0.432 | 0.470 | 0.474 | 0.158 | 0.164 | 0.195 | 0.231 | 0.262 | 0.281 | 0.301 | 0.339 |

## D.3 ABLATION STUDY FOR $L_{gan}$

We provide additional ablation results in Table 12 to evaluate the impact of the loss term $L_{gan}$ in Eq. 3. Performance is measured using both MSE and MAE metrics.

Table 12: Ablation study of $L_{gan}$ on ETTh1. Lower values are better.

| Prediction Length | MSE | | | | MAE | | | |
|---|---|---|---|---|---|---|---|---|
| | 96 | 192 | 336 | 720 | 96 | 192 | 336 | 720 |
| STAR | 0.374 | 0.421 | 0.451 | 0.446 | 0.396 | 0.416 | 0.438 | 0.446 |
| $- L_{gan}$ | 0.378 | 0.428 | 0.454 | 0.452 | 0.399 | 0.418 | 0.442 | 0.451 |

## D.4 PATCHING LENGTH $p$ AND OVERLAPPING TECHNIQUE

Figure 3 presents a comparison of various patch length $p$ under both overlapping and non-overlapping configurations. The results highlight two key findings: (i) the overlapping strategy consistently delivers better performance than the non-overlapping counterpart across all prediction lengths and metrics, suggesting that shared contextual information between patches is beneficial for modeling temporal dependencies; and (ii) among the evaluated patch length, $p = 12$ achieves the lowest error rates, indicating an effective balance between local detail and global context. These observations underscore the importance of patching design in attention-based time series forecasting models.

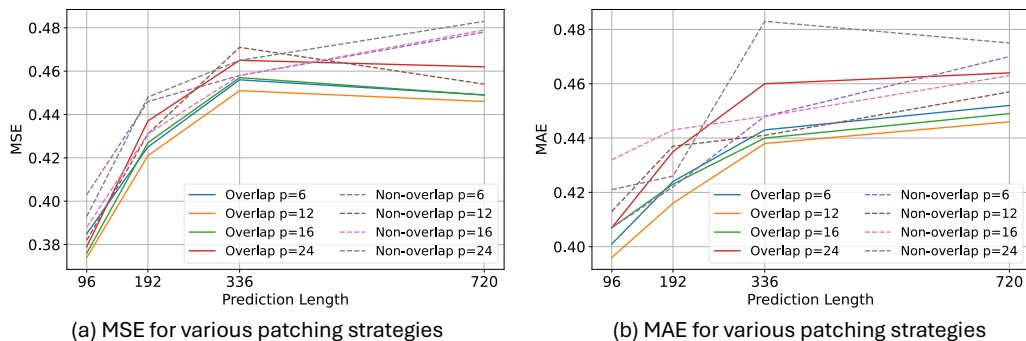

(a) MSE for various patching strategies  (b) MAE for various patching strategies

Figure 3: MSE and MAE for various value of patching length $p$ and overlapping techniques.

# E FURTHER RESULT

## E.1 LLM-BASED COMPARISON

We conducted experiments following the settings of the Chronos Ansari et al. (2024) and Lag-Llama Rasul et al. (2023) and evaluated performance using their **MASE metric** to compare our STAR model with several LLM-based methods. The results, presented in Table 13, indicate that STAR consistently outperforms these approaches across various settings. Note that this comparison is not entirely fair, as LLM-based models typically leverage additional external data for pretraining, whereas STAR relies solely on the provided dataset.

Table 13: Comparison of STAR with LLM-based and other baselines using the MASE metric. Lower is better.

| Dataset | Chronos-T5 (Large) | Chronos-T5 (Base) | Chronos-T5 (Small) | Chronos-GPT2 | Lag-Llama | PatchTST | DLinear | STAR |
|---|---|---|---|---|---|---|---|---|
| Electricity (15 Min.) | 0.077 | 0.078 | 0.080 | 0.082 | 0.319 | 0.082 | 0.079 | **0.072** |
| Electricity (Hourly) | 0.101 | 0.114 | 0.105 | 0.089 | 0.104 | 0.089 | 0.095 | **0.083** |
| Electricity (Weekly) | 0.059 | 0.062 | 0.073 | 0.067 | 0.147 | 0.069 | 0.146 | **0.059** |

## E.2 RESULTS ON DIFFERENT HISTORY LENGTH

We conducted additional experiments with varying history lengths on the ETTh1 and ECL datasets. The results in Table 14 demonstrate that our method consistently outperforms the compared baselines across all history length settings.

Table 14: Performance of different methods under varying input history lengths. Lower values are better.

| Input Length | MSE | | | | MAE | | | |
|---|---|---|---|---|---|---|---|---|
| | STAR | TimeXer | GPT4TS | PatchTST | STAR | TimeXer | GPT4TS | PatchTST |
| 96 | 0.423 | 0.437 | 0.460 | 0.469 | 0.423 | 0.437 | 0.460 | 0.469 |
| 384 | 0.402 | 0.415 | 0.448 | 0.418 | 0.402 | 0.415 | 0.448 | 0.418 |
| 768 | 0.410 | 0.425 | 0.529 | 0.426 | 0.410 | 0.425 | 0.529 | 0.426 |

# F VISUALISATION

In this section, we present visualisations of the predicted time series generated by our STAR model, the diffusion-based ARMD model Gao et al. (2025), and the Transformer-based PatchTST model Liu et al. (2023a). These visualisations reveal the following: (i) The diffusion-based model captures local details well but often fails to model the global shape accurately; (ii) the Transformer-based model, in contrast, captures the global shape but struggles with fine-grained local details; and (iii) by combining the strengths of Transformer architectures and multi-step generation, our STAR model effectively captures both global structure and local detail, demonstrating the key contribution of our work.

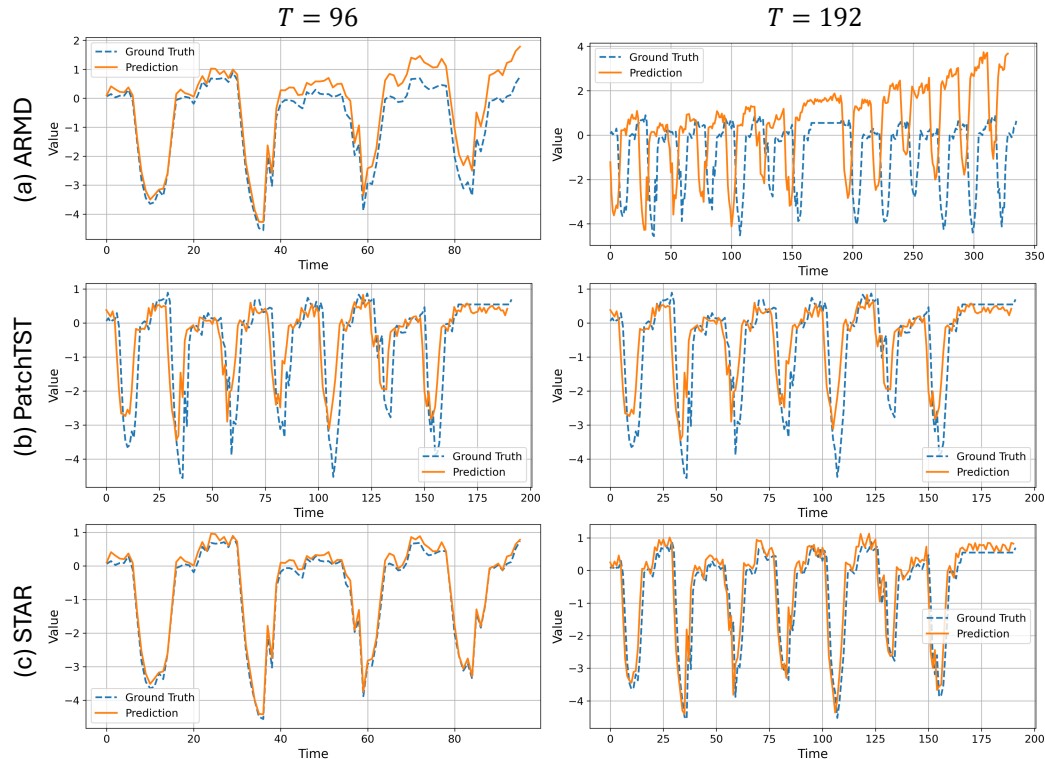

Figure 4: Predicted time series on ETTh1 with forecast horizons of 96 and 192, for (a) the diffusion-based model ARMD Gao et al. (2025), (b) the Transformer-based model Liu et al. (2023a), and (c) our proposed STAR model.

# G  THE USE OF LARGE LANGUAGE MODELS

We used a large language model (ChatGPT) to help with editing this paper. It was only used for simple tasks such as fixing typos, rephrasing sentences for clarity, and improving word choice. All ideas, experiments, and analyses were done by the authors, and the use of LLMs does not affect the reproducibility of our work.

