# OpenReview forum: "STAR: Next-Scaled Autoregressive Model for Time Series Forecasting"
_ICLR.cc/2026/Conference — ICLR 2026 Conference Desk Rejected Submission_

### Official Review · Reviewer_JR1b · 2025-10-16

**Soundness:** 2
**Presentation:** 2
**Contribution:** 1
**Rating:** 2
**Confidence:** 4

**Summary:**

This paper proposes **STAR**, a coarse-to-fine, next-scaled autoregressive framework for time-series forecasting. A Global-Shape Forecaster first predicts a coarse representation (capturing trend/seasonality), then a Next-Scaled Refiner iteratively upsamples/refines to the target horizon. An Inference-Aware Augmentation strategy blends teacher forcing with rollout noise to stabilize multi-step prediction. Experiments on standard LTSF suites (ETT, Weather, ECL, Traffic) and a short-term exogenous setting are reported, with ablations for key components.

**Strengths:**

1. Intuitive coarse-to-fine decomposition; clear motivation for separating global shape from fine-grained refinement.

2. Broad evaluation across common LTSF benchmarks; ablations suggest each component (global stage, refinement, inference-aware training) adds value.

3. Favorable latency vs. diffusion-style iterative methods; method conceptually simple to implement.

**Weaknesses:**

1. As the author mentioned, Next-scale or coarse-to-fine autoregression is well-established in other modalities (e.g., VAR for images, MDLM for text); what is technically new for time series forecasting beyond instantiating the paradigm (masking scheme, conditioning, training losses), and why prior next-scale designs can’t be directly applied.

2. Table 1 does not provide enough support for an SOTA claim. Significant competitive models are missing (e.g., **TQNet** [1], **TimeBridge** [2] (ICML25); **PatchMLP** [3] (AAAI25); **TimeMixer++** [4] (ICLR25)). Given the rapid progress in recent studies, excluding these undermines current SOTA claim.

3. Even against the current listed baselines, improvements are small or dataset-dependent. With multi-seed variance, the gains may vanish. Reporting mean±std over ≥5 seeds and statistical tests are encouraged.

4. It remains unclear whether gains stem from the multi-scale schedule, the masking design, or inference-aware augmentation. Add controlled ablations replacing the custom mask with standard causal attention, removing IAAug, and varying refinement steps K across datasets (not only one).

5. The multi-scale autoencoder uses an adversarial term $L_{gan}$, but the discriminator architecture, objective (hinge/BCE/LSGAN), and stabilization details are omitted; this affects reproducibility and fairness since GAN choices can materially change high-frequency fidelity. Besides, I just found that **the anonymous code link does not open** (access fails).

6. STAR appears to adopt a multi-scale paradigm, yet it does not sufficiently discuss or compare against closely related multi-scale approaches such as **Scaleformer** [5] (ICLR23), **Pathformer** [6] (ICLR24), **PDF** [7] (ICLR24), **mr-Diff** [8] (ICLR24), and **AMD** [9] (AAAI25). Please include a focused discussion of architectural differences (e.g., how STAR’s coarse-to-fine schedule, masking, and conditioning diverge from these) and add quantitative comparisons to substantiate the claimed advantages.

7. Beyond fixed input length $L=96$, please conduct an input-length search (as in DLinear and PatchTST) for all methods to ensure a fair comparison.


*Minor suggestions*:

The quality of the figures could be improved. For example, in **Figure 2**, the formatting of $\mathbf{\hat{r}}_1^x$ is inconsistent in several places, and the label at the bottom, $\mathbf{\hat{r}}_k^y$, is partially occluded.

*[1] Temporal Query Network for Efficient Multivariate Time Series Forecasting.*

*[2] TimeBridge: Non-Stationarity Matters for Long-term Time Series Forecasting.*

*[3] Unlocking the Power of Patch: Patch-Based MLP for Long-Term Time Series Forecasting.*

*[4] TimeMixer++: A General Time Series Pattern Machine for Universal Predictive Analysis.*

*[5] Scaleformer: Iterative Multi-scale Refining Transformers for Time Series Forecasting.*

*[6] Pathformer: Multi-scale Transformers with Adaptive Pathways for Time Series Forecasting.*

*[7] Periodicity decoupling framework for long-term series forecasting.*

*[8] Multi-Resolution Diffusion Models for Time Series Forecasting.*

*[9] Adaptive multi-scale decomposition framework for time series forecasting.*

**Questions:**

See in weakness.

---

### Official Review · Reviewer_dERo · 2025-10-16

**Soundness:** 2
**Presentation:** 1
**Contribution:** 1
**Rating:** 2
**Confidence:** 4

**Summary:**

This paper proposed a model called STAR, which models time series by capturing the features at different scales. It includes two steps: global modeling: Capturing long-term trends and patterns; and local refinement: Adding fine-grained details through progressive generation

**Strengths:**

Experiments strictly following prior work, providing straight forward comparison.

**Weaknesses:**

Method: The method comprises 3 losses. Apart from the standard MSE between predicted and target forecasting values, there is also (1) MSE between the encoded features vs reconstructed features, and (2) a GAN loss. However,  for (1), the ground truth of the feature space is not known, and (2) is also left undefined --- what regularization is applied here and why is needed for the predicted values when they are already constrained by the ground truth $\mathbf{y}$.

Presentation: The notation of the equations and algorithms can be a bit confusing. For example, according to lines 170-172, the feature embedding is obtained from $\mathbf{z}$, but in Eq. (1), the input becomes $\mathbf{y}$. Additionally, some terms / operations are used without explanation. For example, "token map" first appears in lines 173-176. But what the map stores and how the map is established is not explained in the paper. It is unclear to the audience. Is it spatial? Temporal? Or just a tensor dictionary?

In general, the presentation of the paper would benefit from a great revision. I am not convinced that the current version is up to a publishable standard.

**Questions:**

Capturing time series' features at different levels has been a long-discussed topic and has been explored in many prior work (e.g. [1-5]). What makes the proposed method STAR better than the others, and what limitation does this model solve that others cannot?

- [1] TimesNet: Temporal 2D-Variation Modeling for General Time Series Analysis

- [2] Multi-resolution Time-Series Transformer for Long-term Forecasting

- [3] WPMixer: Efficient Multi-Resolution Mixing for Long-Term Time Series Forecasting

- [4] TimeMixer: Decomposable Multiscale Mixing for Time Series Forecasting

- [5] Pathformer: Multi-scale Transformers with Adaptive Pathways for Time Series Forecasting


Why use autoregressive models? Autoregressive models process data iteratively, allowing them to capture strong seasonal patterns present in historical data. While this paradigm works well when the data exhibit consistent seasonality, it may struggle to adapt to sudden changes or regime shifts in dynamic systems. If the data already contains strong seasonal components, what is the advantage of employing more complex deep learning models over traditional autoregressive ones?

---

### Official Review · Reviewer_9Hg8 · 2025-10-24

**Soundness:** 2
**Presentation:** 2
**Contribution:** 2
**Rating:** 4
**Confidence:** 3

**Summary:**

This paper proposes **STAR **, a novel hierarchical forecasting architecture for time series. STAR consists of two synergistic modules:

1. **Global-Shape Forecaster (GSF)**: a Transformer-based component that models coarse-scale, long-term temporal dependencies.
2. **Next-Scaled Refiner (NSR)**: an autoregressive refiner that progressively enhances predictions from coarse to fine temporal scales.

To stabilize autoregressive generation, the authors introduce **Inference-Aware Augmentation (IAAug)**, which simulates inference-time conditions during training to reduce error accumulation.

Experiments on seven benchmark datasets and short-term exogenous tasks demonstrate that STAR consistently outperforms state-of-the-art transformer, diffusion, and linear models across both MSE and MAE metrics. Ablation studies confirm the necessity of each component.

**Strengths:**

1.The coarse-to-fine autoregressive structure elegantly bridges global and local temporal modeling, addressing a long-standing trade-off in time series forecasting.

2.STAR achieves SOTA performance across **multiple domains and horizons**, consistently outperforming both Transformer- and diffusion-based baselines.

3.The paper covers diverse datasets, prediction lengths, ablations, and efficiency analyses.

**Weaknesses:**

1.The paper lacks a formal analysis or justification for why the hierarchical coarse-to-fine paradigm guarantees better generalization or convergence properties.

2.The training involves multiple stages (Autoencoder pretraining, GSF, NSR, and IAAug), which might hinder reproducibility and practical adoption.

**Questions:**

1.While the hierarchical coarse-to-fine paradigm is intuitively appealing, the paper does not provide a theoretical justification for why such a multi-scale refinement guarantees better generalization or convergence properties. In principle, adding more refinement stages may simply increase model capacity or introduce additional global context rather than fundamentally improving optimization dynamics.

Could the authors clarify whether further increasing the number of scales (i.e., deeper hierarchy) continues to yield consistent improvements, and if so, whether there is any theoretical or empirical evidence that supports a convergence or generalization advantage beyond empirical performance gains?



2.The motivation of the Inference-Aware Augmentation (IAAug) module is clear and reasonable — to reduce the gap between training and inference by simulating the model’s behavior during generation. However, the implementation description seems rather heuristic.

Specifically, it appears that IAAug simply interpolates between the ideal input $\tilde{R}$ and a generated input $\hat{R}$ obtained by running the model for several steps, introducing a small perturbation controlled by $\alpha$.
Could the authors clarify **why such a linear interpolation is sufficient** to make the augmented samples closer to the inference distribution? Is there any empirical or theoretical evidence that this augmentation truly aligns the training distribution with the test-time input distribution, rather than merely adding noise?

---

### Official Review · Reviewer_FuD2 · 2025-10-30

**Soundness:** 3
**Presentation:** 2
**Contribution:** 2
**Rating:** 4
**Confidence:** 3

**Summary:**

The paper proposes STAR (Next-Scaled Time-series AutoRegressive Model), which iteratively builds predictions in a coarse-to-fine manner to address the issue that traditional forecasting models often struggle to model both long-term dependencies and fine-grained details of complex temporal data.

The model aims to learn global context and local details in a balanced way via Global-Shape Forecaster (GSF), which predicts the global shape, and Next-Scaled Refiner (NSR), which progressively increases resolution to capture detailed regions.

**Strengths:**

1. The work addresses a practically important problem by explicitly balancing global trends and local patterns via a coarse-to-fine token-map pipeline.

2. The paper demonstrates strong empirical results in accuracy, outperforming state-of-the-art time-series models and LLM-based models across diverse datasets for both short- and long-term forecasting.

**Weaknesses:**

1. While the method builds on known ingredients (coarse-to-fine refinement and a multi-resolution token space), the paper would benefit from a clearer comparison against non-diffusion coarse-to-fine time series baselines such as Scaleformer or C2FAR.

2. Time–complexity trade-off is somewhat less compelling vs single-pass baselines. While much faster than diffusion, multi-step refinement is typically slower than single-pass SOTA.

3. The manuscript provides limited qualitative evidence for the role of each component. In particular, (i) per-scale decoding snapshots for NSR (coarse-to-fine), (ii) token-map visualizations/attention or saliency over time, (iii) case studies and failure analyses across regimes (e.g., spikes, regime shifts). Adding these would clarify each component’s contribution and improve interpretability.

**Questions:**

## A. High-impact questions

1. In Related Work, please provide a clearer comparison with non-diffusion coarse-to-fine time-series baselines (e.g., Scaleformer, C2FAR), detailing architectural, objective, and complexity differences.
2. In Section 3.5, the text reads as Stage-1 for MSTSA and Stage-2 for GSF/NSR, but there is no explicit freeze policy. The manuscript presents the stages as separate, but it is unclear whether AE components are updated in Stage-2. Please clarify what is frozen and what gradients flow where.
3. In Section 3.4, Eq. (12), Each scale $k$ contributes a loss term. Does this implicitly overweight coarser (high-level) scales - since they affect more intermediate reconstructions? Please report an ablation comparing equal weights vs non-uniform scale weights and discuss any differences.
4. If NSR refines across scales, please show qualitative per-step decoding results to illustrate progressive refinement.
5. Beyond per-step decoding, please include token-map visualizations (e.g., attention/saliency over time) and regime-wise case studies/failure analyses (spikes, regime shifts) to clarify each component’s role.

## B. Clarifications that improve reproducibility

1. In Section 3.2, the term 'Residual Quantization' reads like vector quantization. If it is actually a residual decomposition (no codebook/commitment loss), please clarify the terminology or rename accordingly.
2. In Section 3.4, Eq. (8), the RHS product starts at $k=2$, so $r_1^{\mathbb y}$ is not accounted for. Please clarify whether a term is missing, or not.
3. In Appendix D.1, the authors note $p=12$ with overlap so that $T=12$ corresponds to 2 patches. For reproducibility, could you briefly specify the stride (e.g., $p/2$) and any boundary handling such as padding used during patching?
4. In Table 10, which dataset was used for the search in Table 10?
5. In Appendix B, if an LR scheduler is used, could you indicate which scheduler and key hyperparameters?

---

### Author Response · Authors · 2025-12-03

Thanks very much for your comments. Below is my general rebuttal.

**vs other Multi-scale Forecaster:**

Existing methods typically rely on a single-pass Transformer to jointly model multiple forecasting levels (TimeMixer, TimeMixer++, Scaleformer,...). In contrast, STAR is built on a VAR-based autoregressive backbone that generates forecasts iteratively. This iterative refinement enables higher-quality predictions, especially at finer temporal resolutions. As motivated in our paper, the core idea of STAR is to combine the strengths of Transformers (for capturing long-term dependencies and trends) with those of autoregressive generation (for detail refinement). As a result, STAR achieves both stronger long-horizon forecasting and higher-quality local details than prior approaches.

**Why use autoregressive models?**

We adopt autoregressive models because, in our setting, they offer faster iterative generation and stronger performance compared with diffusion-based approaches. Moreover, autoregressive models naturally support multi-scale generation, which is a key strength we exploit in STAR. By combining deep autoregressive modeling with a Transformer backbone, we retain the benefits of classical autoregression (good handling of seasonality and iterative refinement) while gaining the representational power of deep learning to better adapt to complex dynamics and regime changes.

**Compared method?**

There are several forecaster models proposed in recent years. In this paper, we focus on comparing STAR with representative state-of-the-art methods from each major category of time series forecasting. We agree that including additional recent baselines would further strengthen the empirical study, and we will incorporate more comparisons in the revised version of the paper.

**A theoretical justification for why such a multi-scale refinement?**

We view our contribution as primarily empirical. The effectiveness of multi-scale refinement has already been demonstrated in several prior works, and our results further support its benefit in the context of long-horizon time series forecasting. A full theoretical analysis of multi-scale refinement would require substantial additional assumptions and space, and is therefore beyond the scope of this paper. We consider this an interesting and important direction for future work.

---

### Note · Program_Chairs · 2026-01-17
**Submission Desk Rejected by Program Chairs**

The following references in this submission do not refer to real documents and/or have major errors in bibliographic information:

 Ailing Zeng and et al. Rlinear: Revisiting linear transformers for time-series forecasting. In International Conference on Machine Learning (ICML), 2023.